# A Study on Industrial IoT for the Mining Industry: Synthesized Architecture and Open Research Directions

**Abdullah Aziz \*, Olov Schelén and Ulf Bodin**

Department of Computer Science, Electrical and Space Engineering, Luleå University of Technology, 971 87 Luleå, Sweden; Olov.Schelen@ltu.se (O.S.); Ulf.Bodin@ltu.se (U.B.)

\* Correspondence: abdullah.aziz@ltu.se

**Abstract:** The Industrial Internet of Things (IIoT) has the potential to improve the production and business processes by enabling the extraction of valuable information from industrial processes. The mining industry, however, is rather traditional and somewhat slow to change due to infrastructural limitations in communication, data management, storage, and exchange of information. Most research efforts so far on applying IIoT in the mining industry focus on specific concerns such as ventilation monitoring, accident analysis, fleet and personnel management, tailing dam monitoring, and pre-alarm system while an overall IIoT architecture suitable for the general conditions in the mining industry is still missing. This article analyzes the current state of Information Technology in the mining sector and identifies a major challenge of vertical fragmentation due to the technological variety of various systems and devices offered by different vendors, preventing interoperability, data distribution, and the exchange of information securely between devices and systems. Based on guidelines and practices from the major IIoT standards, a high-level IIoT architecture suitable for the mining industry is then synthesized and presented, addressing the identified challenges and enabling smart mines by automation, interoperable systems, data distribution, and real-time visibility of the mining status. Remote controlling, data processing, and interoperability techniques of the architecture evolve all stages of mining from prospecting to reclamation. The adoption of such IIoT architecture in the mining industry offers safer mine site for workers, predictable mining operations, interoperable environment for both traditional and modern systems and devices, automation to reduce human intervention, and enables underground surveillance by converging operational technology (OT) and information technology (IT). Significant open research challenges and directions are also studied and identified in this paper, such as mobility management, scalability, virtualization at the IIoT edge, and digital twins.

**Keywords:** IIoT; mining industry; industrial standards; industry 4.0; industrial edge computing; edge virtualization

---

## 1. Introduction

The Internet of Things (IoT) is a technological paradigm imagined as a global network where devices or machines can interact [1]. IoT is acting as a technological revolution influencing all application domains including smart home, smart cities, agriculture, automobiles, health-care, industrial production, and transport [2,3]. It is estimated that there will be 50 to 100 billion smart things and objects connected to the Internet by 2020 [4–6]. In this context, industries are being challenged to rethink their production processes with the potential to spark innovations in production systems on an unprecedented scale [7].

The Industrial Internet of Things (IIoT), which is an application of IoT in industry, is part of the Industry 4.0 concept, which emphasizes the idea of consistent digitization and the connectivity of all productive units [8], combining the strengths of the traditional industry with internet technologies [9,10]. The Industrial Internet may also be considered as a convergence of Information Technology (IT) and Operational Technology (OT) as shown in Figure 1. IIoT is the network of physical objects, or things, embedded with electronics, sensors, and connectivity to enable that network to achieve greater value and service by exchanging data with the manufacturer, operator, and/or other connected devices [11]. Currently, many IoT technologies are integrated into consumer applications, such as smart homes, connected cars, and smart wearables. The industrial applications of IoT, or Industrial IoT (IIoT), however, are anticipated to have the capability to transform many industries, including manufacturing, oil and gas, agriculture, and mining [12].

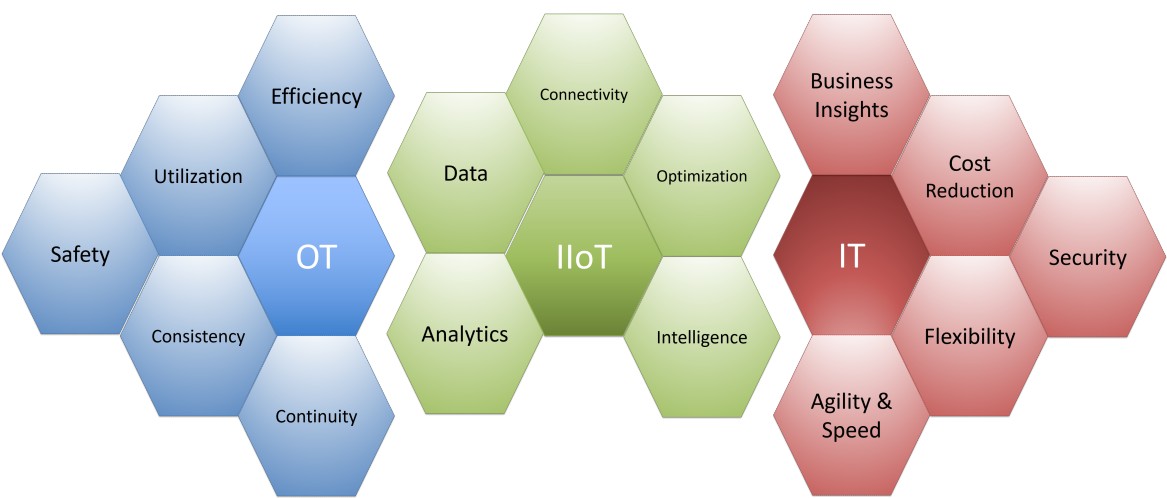

**Figure 1.** Operational technology (OT) and information technology (IT) = Industrial Internet of Things (IIoT).

The scope of this research is the adoption of IIoT in the mining industry, specifically in the case of underground mining. The complexity and specific challenges in this industry include: heterogeneity where each mine has a different layout depending on the natural conditions, heavy transportation in combination with very confined spaces, wireless communication across shafts and stopes not possible, repeated blasting where devices would be destroyed unless moved every time, face drilling that changes the layout continuously, the need for fail-over and autonomous operations to ensure production and safety through catastrophic failures, etc. These challenges are further accentuated by the need for continuous production of large volumes of extracted ore to reached targeted goals set to ensure profitability in production. Other types of industries may be able to catch up after a stop in production, while in mining lost production can typically not be recovered by an increased phase of operations. The unique combination of these conditions is a motivation for the study. In particular, it is motivated by the complexity in terms of the need for flexible and adaptable IIoT systems to cope with the continuous change in topology and use of equipment and machines combined with the need to ensure production and safety although connectivity may occasionally fail.

In this paper, we study the current practices of information technology (IT) in the mining industry and find a major challenge of interoperability between various systems and devices. Interoperability [13] is a characteristic in an architecture where different systems or devices can communicate with other systems and devices to exchange, understand, and make use of the information. The article also presents the study of the architectural models and features of some generic IIoT frameworks, synthesizes them in the perspective of requirements from the mining industry, and identifies the futuristic challenges and open issues. With the advancements in information technology, the mining industry is increasing its adoption of digital and technology solutions.

The primary purpose of this adoption is to achieve cost and productivity based optimal solutions, increase safety measures, and developing their intelligent systems. Mostly mining companies are gathering data by the use of sensors and mobile telemetry to facilitate operational managers to improve operational efficiency. These adoptions are mainly carrying by individual mining companies to fulfill their specific needs and not well standardized for global adoption in the mining industry. The operations and workflow of the mining industry are different than other industries such as manufacturing, agriculture, and transportation industries, etc.

Boliden AB [14] is a Swedish mining and smelting company focusing on the production of copper, zinc, lead, gold, and silver. They are among the leaders in the digitalization [15] of mining operations and are therefore representative as a base for state-of-the-art. The overall strategy of Boliden is to create profitable growth adapting to recent trends in the market and investments in competitiveness and organic growth; and by evaluating opportunities for acquisitions. For data derived processes the insights and decisions are getting increasingly important. Many mining sites have in pockets highly mature proprietary systems, but when it comes to interoperability with different systems or information sharing among departments or stakeholders, analytics capabilities, modern IIoT practices, and architecture, there is need for improvements. A strategy has been set and several projects are ongoing such as reporting and visualization of processes data, setting up a production data platform for mines, ideas on a common data platform for smelters as well as activities coordinate by the automation's programs. Historically, Boliden is operating in a decentralized model, and in the reporting and analytics field, the business areas and sites have started loosely coordinated activities. In addition, to obtain extendable and scalable systems, there is a need to analyze the industrial standards and practices and put them into a mining industry perspective.

The industrial IoT standards and initiatives are evolving and many industries are getting benefits (e.g., increase efficiency, reduce errors, predictive maintenance, improve safety, and reduce cost) [16] from the industrial IoT standards and guidelines such as the manufacturing industry [17,18], healthcare, smart cities [19], and transportation [20]. The focus of this article is to investigate the possibility to apply IIoT in the mining industry whether these standards and initiatives give guidelines that can benefit the mining industry. Moreover, this study shows how these standard guidelines can help to address the current challenges in the mining industry such as the interoperability problem due to vertical fragmentation, data distribution, and the exchange of information securely.

The remaining sections of the paper can be grouped as background study (Sections 3 and 4), related work of IoT in the mining industry in Section 5, and our contributions (Sections 6–9). This article targets the readers from the two different domains of information technology and mining engineering therefore covers the background study for both. A reader from mining engineering who knows the mining life cycle can skip through Section 3. Similarly, a reader from the information technology domain who already knows the listed IIoT standards and related initiatives can skip through Section 4. Whereas the main contributions are as follows:

- We describe the current IT practices in the mining industry and identify the challenges (Section 6).
- We use the guidelines and key considerations from the industrial IoT standards and initiatives to synthesize a high-level IIoT architecture for the mining industry (Section 7), and the reflections of the architecture (Section 8).
- Finally, we analyze and identify some futuristic IIoT challenges and some possible future work directions (Section 9), and conclude the manuscript (Section 10).

## 2. Research Methodology

Our research aims to improve the adoption of IIoT in the mining industry by synthesizing a high-level architecture, using the guidelines of the industrial standards, and meeting the specific challenges in the mining industry. Thus, qualitative synthesis was used as the research methodology [21,22].

Boliden is a leading Swedish mining and smelting company and known to be in the forefront of digitization and automation of mining [15]. Therefore, to study the present use of IIoT in mining, we chose to work with them. The IT team of Boliden demonstrated their current IT practices (Section 6) where standalone commercial solutions and systems operate in isolation without interworking with other systems. Therefore, manual information exchange between those systems by human intervention is a common practice.

To address these challenges we performed a literature study (Section 5) and found out that there are various IoT solutions in the mining industry that are specific to one problem, forming vertical silos in the mining industry. Further investigation to this problem revealed that the existing IoT solutions are not following the standard guidelines, and to make them interoperable there is a need for a standard-based IIoT architecture for the mining industry.

Our research selected a set of industrial standards (Section 4) providing industrial automation, interoperability between different legacy and modern systems, real-time analytics at the edge for continuous and safe mining operations, business analytics, distributed data management across various mine sites, information exchange between departments and stakeholders, and security. A synthesized high-level architecture is then presented (Section 7) which addresses the identified challenges of the mining industry.

## 3. Introduction to Mining: Stages in the Mining Life Cycle

In this section, we introduce the stages of the mining life cycle to show the operations in the mining industry. As shown in Figure 2, the overall sequence of activities in modern mining is often compared with the five stages in the life of a mine: prospecting, exploration, development, exploitation, and reclamation [23].

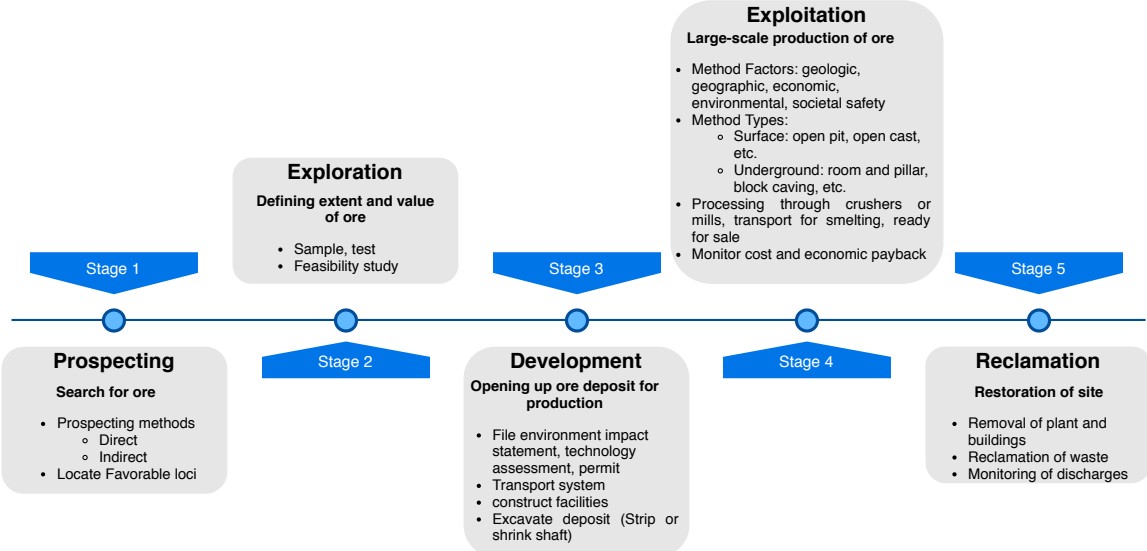

**Figure 2.** Stages in the mining lifecycle.

### 3.1. Prospecting

Prospecting, the first stage in the utilization of a mineral deposit is the search for ores or other valuable minerals (coal or nonmetallic). Because mineral deposits may be located either at or below the surface of the earth, both direct and indirect prospecting techniques are employed.

*3.2. Exploration*

The second stage in the life of a mine, exploration, determines as accurately as possible the size and value of a mineral deposit, utilizing techniques similar to but more refined than those used in prospecting.

*3.3. Development*

In the third stage, development, the work of opening a mineral deposit for exploration is performed. With it begins the actual mining of the deposit, now called the ore.

*3.4. Exploitation*

Exploitation, the fourth stage of mining, is associated with the actual recovery of minerals from the earth in quantity. Although development may continue the emphasis in the production stage is on production. Usually, only enough development is done before exploitation to ensure that production, once started, can continue uninterrupted throughout the life of the mine.

*3.5. Reclamation*

The final stage in the operation of most mines is reclamation, the process of closing a mine and re-contouring, re-vegetating, and restoring the water and land values. The best time to begin the reclamation process of a mine is before the first excavations are initiated. In other words, mine planning engineers should plan the mine so that the reclamation process is considered and the overall cost of mining plus reclamation is minimized, not just the cost of mining itself.

## 4. Industrial IoT Standardization and Related Initiatives

In this section, we cover relevant industrial reference architectures which are proposed by the standard bodies.

*4.1. IIRA*

Industrial Internet Reference Architecture (IIRA) [24] started in 2014 and gives guidelines to construct industrial internet systems. It describes a standard open architecture framework that helps to design industrial internet systems with modern capabilities. It characterizes conventional architectural concerns and organizes them into the four viewpoints: business, usage, functional, and implementation. IIRA states that the functional domain is the most important part to consider for an IIoT system which describes five-layers as follows:

- Business layer: this layer provides the functions to allow end-to-end industrial operations such as work planning, scheduling, enterprise resource planning(ERP), and life cycle management.
- Application layer: it provides the functions to allow the execution of some task or goal by implementing the actual application logic such as application programming interface, and user interface, etc.
- Information layer: this layer offers functionalities to collect and store data and semantics, transforming and analyzing data, data provisioning, and deployment.
- Operations layer: this layer provides functions for assets to operate properly during the life-cycle. It is also responsible for asset deployment, configuration, diagnosis, and update.
- Control Layer: functions of this layer provides the control of industrial assets such as sensors, actuators, and communication.

*4.2. RAMI 4.0*

RAMI 4.0 is a three-dimensional six-layer architecture which is proposed by the German Plattform Industrie 4.0 and provides a service-oriented-architecture (SOA) framework for interoperability to assist and stimulate the main characteristics of industry 4.0. The main focus of RAMI 4.0 is the integration of

industrial assets from the shop floor to the office floor with SOA-based services and applications in the manufacturing industry. It does not provide a detailed methodology of technical implementation but identifies the manufacturing industry standards that are analyzed in [25]. The core specifications of RAMI 4.0 was published in [26] and applied to the field of linked data in [27]. Both RAMI 4.0 and IIC's IIRA provide reference architectures for industrial systems and the comparison of both architectures is published in the joint whitepaper [28].

*4.3. oneM2M*

oneM2M [29,30] was formed in 2012 and it is an initiative of eight leading ICT standards organization: European Telecommunications Standards Institute (ETSI) from Europe; Association of Radio Industries and Businesses (ARIB) and Telecommunication Technology Committee (TTC) from Japan; China Communications Standards Association (CCSA) from China; Alliance for Telecommunications Industry Solutions (ATIS); Telecommunications Standards Development Society (TSDSI) from India and Telecommunications Industry Association (TIA) from the USA; and Telecommunications Technology Association (TTA) from Korea. The goal of oneM2M is to define a horizontal service layer [31] to provide a way to connect and communicate with different IoT systems. Different vendors offer IoT systems unable to inter-work with other IoT systems and forms vertical pipes causing fragmentation in the market. The oneM2M horizontal layered architecture [32] addressing this problem by reducing the vertical fragmentation in the market [33]. A joint paper [34] was written by IIC and oneM2M which maps the IIRA and oneM2M in detail and provides the future directions for both organizations.

*4.4. Arrowhead Framework*

The EU Arrowhead Project [35] started in 2013 and proposed an architecture [36] to describe the workflow of IoT-based automation. The proposed architecture and open-source framework is according to the guidelines of RAMI 4.0 [37]. The project is already underway in the context of the successors Productive 4.0 [38] and Arrowhead Tools [39] projects. These projects continue the work on the Arrowhead Framework [40] aiming to facilitate the development, deployment, and orchestration of integrated systems based on the SOA concept in an appropriate manner. The Arrowhead Framework forms SOA based Local clouds with three mandatory core systems which are Service Registry, Authorization, and Orchestration system. To enhance the capabilities of the Arrowhead Local Clouds, there are several supporting systems and services are available such as EventHandler, Gateway, Gatekeeper, QoS managing, and monitoring system and many are under development for the future. The further work on Arrowhead Framework is done in the Eclipse Arrowhead project [41]

## 5. Related Work

There are only a few academic efforts and studies available to apply IoT and modern technology in the mining industry. In [42], researchers investigated IIoT applications to identify the challenges of the mining industry and determined whether an underground mine is capable of supporting IIoT systems or not. A review of digital transformation in mining [43] presented the foundational parts as ubiquitous data, connectivity, and decision making. An IIoT and Advanced Analytics framework is proposed in [44], it provides a layered architecture for analytics to be used as a guide and facilitator for the adoption of IIoT in the mining industry.

Various research efforts tried to solve specific problems in the mining industry by introducing IoT such as gas monitoring system [45] and employee positioning system [46] in the coal mine. Other pieces of work tailors dam monitoring and pre-alarm system in mines [47], tracking of equipment for maintenance [48], improving machine safety [49], accident analysis system [50], oxygen concentration system [51], fleet and personnel management system [52], ventilation monitoring system [53], and underground mine air quality pollutant prediction system [54].

The IoT architecture and technology is presented in [55] to enable the creation of a digital mining platform highlighting the exploration of rock–fluid–environment, new opportunities for IoT implementations are offered in discovering new mineral infrastructure, better tailings disposal, tracking and prevention of mining pollution. Explored also are the cutting edge innovations which could be leveraged to build the state-of-the-art sustainable IoT mining model.

Many commercial solutions are available in the market for the mining industry and most of which are discussed and evaluated in [56]. Moreover, Ref. [57] also described the application of industry 4.0 in the mining industry by the Software Development Life Cycle (SDLC) perspective and introduces the semi-smart mine, while only focusing on industrial integration paradigms. An IP-based multimodel sensing platform for realizing the vision of IoT in underground coal mines is developed and described in [58]. It is a Zigbee-based wireless sensor network (WSN) initially established and extended to IoT with IP enabled gateway.

The academic literature clearly shows that the pace of implementation of industrial internet in the mining industry is slower. In addition, the solutions to solve specific problems shown in the literature do create a challenge of interoperability between various solutions. There is a need for an IIoT architecture for the mining industry that allows the smooth interoperability to exchange information from the mine sites towards the operational units and office floor. Moreover, the architecture should follow the guidelines of the industrial standards designed by standard bodies and adopted by a large number of vendors. To address this research gap, this study synthesizes an IIoT architecture by using the guidelines of the industrial standards for the mining industry.

## 6. Current IT Practices in Mining Industry and Key Challenges

In this section, we describe the current IT practices after studying the IT infrastructure in the mining companies. Though not under the term IIoT, the mining companies have been deploying programmable logic controller (PLC) and supervisory control and data acquisition (SCADA) systems for monitoring and controlling for decades e.g., there exists a number of commercially available systems. Compared with IIoT systems, these existing monitoring and control systems are generally proprietary systems and were not designed to interoperate or interconnect with other systems. The major difference between IIoT-based systems and legacy systems is that IIoT systems are based on an open, highly connected Internet Protocol (IP) network structure.

Currently, the mining industry is highly dependant on commercial systems and applications which causes vertical fragmentation. Different stages in the life of mine are described in Section 2. Exploration is the starting point where geologists take samples from specific locations and test in the laboratory. These samples are then analyzed based on various factors such as the size of holes and materials extracted from the holes. After analyzing the samples, geologists create Geo-statistical models. This department of exploration has its own tools and software. The final results are then handed over to the planning department. Planning departments also have their own set of software and tools to make plans based on the input from exploration results. The problem here is that the exploration department export results in an excel sheet to share with the planning department where the results can be only import from the excel sheet because there is no communication established between these systems. Similarly, the further processes are also acting as independent and have no integration with each other such as rock mechanics, drilling and blast, load and haul, crushers, stockpiles, mills for processing, and concentration. Finally, the material is stored in the warehouses and the sales department needs to sell and manage the records.

These various departments based on different stages of the mine life are using different IT systems design and developed by different vendors. The popular commercial systems in the industry are ABB, OSIsoft, IBM Maximo, Microsoft Azure, Dynamics 365, and Power BI. IBM Maximo offers solutions for asset management, ABB provides the production solutions and popular within control systems. Some IoT devices are attached to ABB systems, which carry vital information. The OSIsoft is trying to get data from various systems to integrate, but most of the data are integrated via excel sheets.

The analytics and machine learning components are also needed for better operations and benefits for the mining industry. For analytics Microsoft Azure is in use which gets data from the data lake, OSIsoft, ABB by importing excel sheets.

As shown in Figure 3, all these systems, and applications are not able to interoperate. These commercial systems have their individual technology stack and data formats, hence one system can interoperate with another system of the same provider but unable to interoperate with a different provider. This causes a big hurdle for applying industrial internet paradigms in the mining industry and slows production.

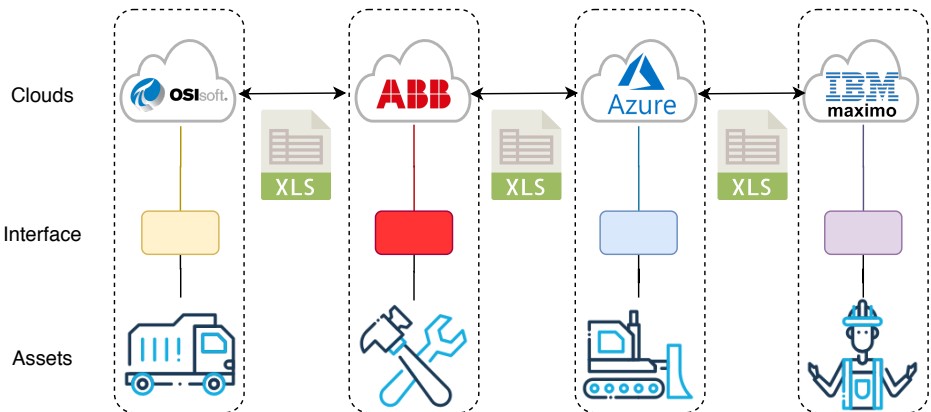

**Figure 3.** Vertical silos in each domain of mining.

There are various challenges to achieve interoperability and communication between different systems in the mining industry. As shown in Figure 3 the current state of the systems in the mining industry is as vertical pipes. The mining industry is facing various key challenges which are summarized as follows:

- Technological Variety: the development of IIoT based system is based on five main technologies: wireless sensor network (WSN), radio frequency identification (RFID), cloud computing, middleware/gateways, and IIoT application software [1]. Moreover, there are various already deployed systems that are based on a different set of technologies.
- Possession of data: the matter of data possession or the ownership of data is an important challenge not only in the mining industry but across many other industries. Industries usually tend to keep the data as well as the information to interpret the data. The owner is typically the one who generates the data, but the system provider collects the data from various customers and can identify the trends by analyzing it.
- Data distribution between legacy systems: in the mining industry, data is distributed across different legacy platforms and systems that causing problems to gather the information for analytics and further operations.
- Security: IIoT enables various systems and applications to communicate with each other and open up various security threats. The communication channels should be secure, IIoT devices become more vulnerable to various attacks such as DDoS. Moreover, the rules for data governance and specific security planning is needed.
- Data Management: In the IIoT environment, devices and assets produce data which then store into the central data layer which later use by analytics applications. There are various data sources in the mining industry such as Enterprise Resource Planning (ERP) system, PIMS, Manufacturing Execution Systems (MES), and many legacies IT systems. In this scenario, data management is a challenging task that requires the use of data lakes, migration, master data, and replication. This challenge becomes worst when there are different cloud providers involved, which is the typical case in the mining industry.

- Analytics: the analytics in the mining industry have added importance compared to the other industries because of the heavy machinery, human force, mill functions, and stockpiles. The operation managers need better insights into the processes and machines, so they can detect issues and act accordingly. There is also a need for edge analytics at the mine sites which targets mostly the operations, but analytics also needs on the office floor for business trends and needs.

Moreover, all of the industrial standards and reference architectures described in Section 4 are trying to address the challenges in industries in general. These standards are providing the guidelines and best practices to design and develop a solution for a specific use-case. Certainly, we cannot directly apply these standards and reference architecture to fulfill the requirements in the mining industry, although these standards can guide us to design an architecture for the mining industry.

## 7. A High Level IIoT Architecture for Mining Industry

In this section, we synthesize a high-level architecture according to the global organization's practices and standards guidelines. The summary of the key considerations taken from the industrial standards to synthesize IIoT architecture for mining is described in Table 1. In the architecture, we have considered the IIRA's viewpoints, functional domains, and crosscutting functions, which is providing rich guidelines for the industry and we have taken those considerations to define different domains of the IIoT architecture for the mining industry. RAMI 4.0 reference architecture also gives guidelines for the mining industry layered architecture and cross-cutting functions. The administration shell of RAMI 4.0 is considered for control and operation domains, where the tasks in the control domain of the mine site are managing from the operation domain. Moreover, the oneM2M standard offers a common service layer to provide a horizontal view for vertical silos. The oneM2M CSE approach is considered in this architecture as an information domain together with various service functionalities such as group management, subscription and notification, data management and repository, and security. Lastly, for the mine sites, we considered an approach of Arrowhead Framework based local cloud automation where various systems can register and provides services. These services can interoperate with each other securely and can be discovered and orchestrated by the core components of the local cloud.

**Table 1.** Key considerations from IIoT frameworks and standards.

| | Key Considerations |
|---|---|
| IIRA | IIRA's viewpoints to investigate into mining industry. IIRA's functional domains to categories the domains in the mining industry. IIRA's crosscutting functions necessary for the mining industry. |
| RAMI 4.0 | Six-layers of RAMI 4.0 guidelines to investigate into mining industry layered architecture approach. The crosscutting functions necessary for the mining industry. The administration shell of RAMI 4.0 to categorize control and operation domain. |
| oneM2M | oneM2M's common service entity as information domain. Data management & Repository for information domain. Group management for information domain in the mining industry for different departments & stakeholders. Subscription and Notification for information domain. Security consideration for information domain. |
| Arrowhead Framework | The concept of local clouds for the edge of mining sites. Security consideration for the automation at the control and operation domain. The orchestration mechanism at the control and operation domain. |

The layered architecture is shown in Figure 4 and divided into five domains. Each mine site consists of two domains such as the edge control domain and the edge operational domain. Information and analytics domains can also deploy on-site in a single mine site at the edge or off-site at the cloud. The off-site information domain at the cloud is connected to all mine sites and domains above that. Furthermore, the analytics domain, business and application domain reside on top of the information domain. The flow of information, commands/requests, and decisions are also shown in Figure 4 with different arrows. The detailed descriptions of each domain are as follows.

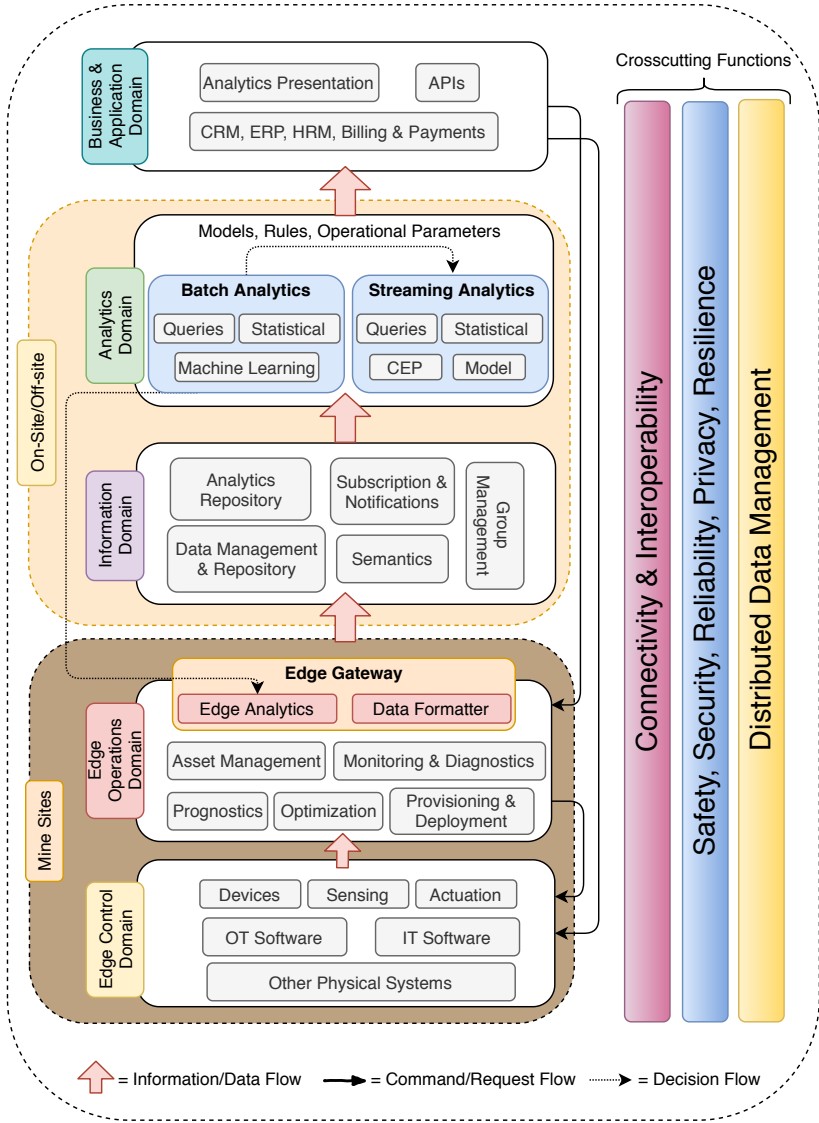

**Figure 4.** Synthesized high level IIoT architecture for mining industry.

## 7.1. Edge Control Domain

The control domain in the mine site consists of various components such as OT software, IT software, sensors, actuators, devices (mobiles), and other physical systems. This domain is responsible for various functions that are carried out by industrial control and automation systems. These functions involve reading data from sensors, applying rules and logic, and send commands to actuators to control the physical systems. The control domain in the mining site usually requires high timing accuracy. The components or devices that perform such tasks in the control domain are typically located in near proximity to the physical systems they control and can also be dispersed geographically. The maintenance staff may not be able to access these devices and physical systems easily after deploying them in the mine site.

## 7.2. Edge Operation Domain

The mines are of different types such as surface and underground mines, therefore the operation domain can be inside the mine or on the surface at the start of the mine in the case of the underground mine. The operation domain is responsible for the management and operation of the control domain. The functions in this domain are in charge of provisioning and deployment, asset management, monitoring and diagnosis, optimization, and prognostics.

Provisioning and Deployment comprise a collection of functions required for the configuration, onboarding, registration, and monitoring of assets and the deployment and retirement of assets.

Asset management consists of a set of functions allowing assets management centers or systems to send commands to the control systems, then from the control systems to the assets. These commands are bi-directional so the assets can respond to these commands back to the asset management centers.

Monitoring and diagnostics are responsible for the detection and prediction of problem occurrences. This is responsible for real-time analysis of asset core performance metrics, intelligence-based gathering, and processing of asset safety data so that it can detect the true source of a problem, and then warn about suspicious conditions and anomalies.

In mining operations, prognostics are really important and comprise the collection of tasks that the IIoT systems provide as a predictive analytics driver. It relies on historical evidence about asset activity and efficiency, asset property in engineering and physics, and knowledge about modeling.

Optimization consists of a series of functions that enhance the efficiency and performance of assets, minimize energy usage, and maximize the quality and production in accordance with how the assets are used. By recognizing output gaps and inefficiencies it helps maintain assets operate at their peak performance.

Furthermore, at the mine site, there is an edge gateway that is responsible for real-time edge analytics and the data formatter. The edge gateway is also responsible for the connectivity and interoperability between various vertical solutions, which we will discuss later in this paper. The key thing to mention here is that the data formatter will convert all the data from the operation and control domain in a form understandable for the information domain.

## 7.3. Information Domain

The information domain is responsible for managing and processing data. This domain acts as an information hub with the functions for gathering data from various domains most importantly from control and operation domains and can reside on-site (locally in the mine) or off-site (on cloud). In the mining industry, this domain is the centralized data repository that stores all the data from various departmental systems including exploration, geologists, crushers and mills, waste management, sales, and business.

Data Management and Repository includes the ability to collect data to aggregate large amounts of data, convert that data into a specified format, and store it for other purposes such as analytics and semantic processing.

The information domain also provides Group Management which is responsible for handling group-related requests. The users from various departments or stakeholders can make different groups to have appropriate access to the information domain. With the help of group management in the information domain, we can also achieve horizontal and end-to-end integration by providing access to the information with group-based credentials.

In addition to using group management or other traditional access control models such as Mandatory Access Control (MAC), Discretionary Access Control (DAC), and Role-Based Access Control (RBAC), Attribute-Based Access Control (ABAC) may be considered. With ABAC high authorization granularity, central administration of access policies with centrally consolidated and monitored logging properties are achieved, which can provide a more powerful model for access control in industrial automation systems [59].

Subscription and Notification provide notifications based on the subscription that tracks updates on a specific event. Analytics Repository is a database of raw and harmonized data from multiple sources. The analytics repository transforms data in a way so that it can provide the data to the analytics domain for enterprise-level analytics.

The semantics have the semantic data for the resources which can be used for semantic interoperability between various stakeholders. Although there is a need for specific ontology and

common grammar so the participating systems can decide upon an ontology model to communicate with each other.

## 7.4. Analytics Domain

The analytics domain is responsible to get data from the information domain, mainly from the analytics repository, and perform analytics. Similar to the information domain, it can also reside on-site (locally in the mine) or off-site (on cloud). The analytics in the mining industry has challenges as the results in the physical world can alter the operation and safety of things. Such results can be unintended or dangerous and may unintentionally impact people's health or damage property and the atmosphere. The analytics domain in the mining industry can use both batch and stream analytics processing models. The analysis of batch data is an incredibly effective method of storing vast quantities of data that are gathered over time. Stream analytics is the process of being able to analyze data that streams from one device to another almost instantaneously.

## 7.5. Business and Application Domain

The business and application area is a technical environment for the execution of practical enterprise and application logic. This reflects business functions that help business processes and the business functions of administrative tasks that an IIoT framework will implement to facilitate end-to-end operations. Various business functions include ERP, CRM, HRM, Billing, and Payments. Analytics presentation consumes data from the information domain and analytics domain for report generation and analysis via tools that can be used for ad-hoc queries, reports, and dashboards.

## 7.6. Crosscutting Functions

The various domains described above focuses on major system functions that are generally required to support IIoT implementation in the mining industry. In order to allow the main system functions, additional functionality must be given. These supporting features, the so-called crosscutting functions, also need to be made accessible across several practical components of the architecture which are as follows:

### 7.6.1. Connectivity and Interoperability

Connectivity and interoperability are important for IIoT systems. In the case of the mining industry, these issues are challenging because of vertical silos and legacy systems. The synthesized architecture offers a connectivity and interoperability cross-cutting function according to two main approaches as shown in Figure 5. This is to ensure that heterogeneous systems can communicate.

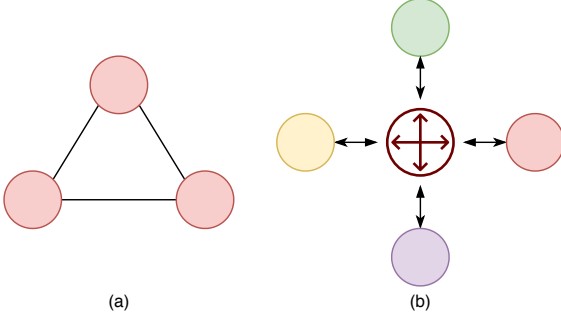

**Figure 5.** Interoperability models: (**a**) common-meta model, (**b**) broker based.

- Broker Based: the broker method as shown in Figure 5b, analogous to utilizing a translator between two separate language speakers. The broker approach works fairly well with a limited number of communicating parties. Perhaps it is the best way to allow communication between parties, where they were previously designed with separate requirements and now feel the need to

interact. Moreover, a broker-based approach is also feasible for sharing the data where one system only wants some specific information from another system. In this scenario, a PubSub based broker can act as a middle-ware between these systems, and whenever the event or data occurs in a system the broker notifies the other system who is interested in that data and already subscribed.

- Common Meta-Model: the common meta-model approach as shown in Figure 5a, needs foresight for architecture interoperability as the communicating parties must be produced using the standard meta-model and interfaces decided upon. There are several directions to achieve common meta-models and agreed-upon interfaces. The predominant strategies include common specification, common modules, standard-based open-source built, open-source frameworks, and closed ecosystem framework.

In the context of the mining industry, Boliden is also facing the same problem of vertical silos. To overcome the hurdle of data sharing between different verticals Boliden designs a solution based on a broker-based interoperability model. An MQTT-based broker is designed and placed in between the already deployed systems by various vendors. The reason for choosing the MQTT as the middle-ware is that it is open source and has a rich inventory of libraries to work with different technologies. There are specific topics registered in the broker and the source systems publish data to the topic in the broker. On the other hand, all the listeners subscribe to the specific topic, and whenever a new event comes broker broadcast the data to all the listeners. Moreover, the broker also implemented a data formatter that translates the data according to the source and destination platforms.

### 7.6.2. Distributed Data Management (DDM)

The data is distributed between various mine sites and edges, therefore to tackle the challenge of managing the data produced by various devices and applications in different mine sites, DMM is also considered as the cross-cutting function. DMM involves three steps: (1) data collection, (2) data aggregation, and (3) storage, whereas the connectivity is a prerequisite function. Moreover, DMM contains some systematic tasks such as:

- Publish and subscribe: publish and subscribe is ideal for sharing data notifications across components that are loosely coupled and enable the publish–subscribe model to customize the communication route across publishers and subscribers depending on their needs. It supports different kinds of data such as streaming data, alarm and event, command and control, and configuration data. It gives a reliable data flow from the edge devices to the central data storage, scalable handling of a large proportion of increasingly data sources and consumers, and reliable control commands flow from applications or management services to devices.
- Reduction and analytics: IIoT devices and applications generate large amounts of data and transmitting all this raw data to the central data center are often pointless and expensive, but the insights hidden in the raw data must not be lost. Reduction and analytics may handle data by either reducing the quantity or pace without compromising the hidden insights of the data.
- Query: DMM makes queries using two models: (1) the one-time query models are based on request-response pattern and well suited for traditional databases, (2) the continuous query model is compatible with the in-memory databases and data stream management systems. To select a subset of data from a larger set, DMM uses a combination of two styles: save data; run query and its reverse, save query; run data. Both query styles and models may implement at different levels.
- Storage, oersistence, and retrieval: storing and persisting subset of data in sequential order with time-stamping helps to order identification and replication across various datasets. It helps to create audit records for future auditing, simulations and various forms of testing, and reliable storage and scalable archiving.

### 7.6.3. Safety, Security, Reliability, Privacy and Resilience

To ensure security in the system-wide components, a certain collection of security functions, as crosscutting functions, must be incorporated in each of the operational components and their

communications, such as encryption and authentication. The overall security of an IIoT system of the mining industry is based on how these systems have implemented the security and how securely they are integrated. As a whole, the secure and trustworthy system depends on a collection of certain features such as.

- Safety: safety is the state of the system functioning without creating an inappropriate possibility of human injury to people working in the mines. Traditional OT safety-assessment approaches concentrate on physical objects and procedures and then integrate the chances of calculated element failure into overall system risk. Risk analysis to detect risks aims at avoiding erroneous procedures and enhancing system tolerance to unpredictable incidents. Whereas a software component behaves exactly as programmed, but adversaries can exploit security-related systematic vulnerabilities. A remote attacker is able to exploit weaknesses in the connected systems of mines to drive a device into an insecure state e.g., an autonomous vehicle, so it is important to use safety regulations and practices.
- Security: the security of the system is a continuous behavior and not a Boolean state. No IIoT system can continue to act securely in any scenario so it is necessary to clearly indicate the specific circumstances considered important along with the secure behavior that is expected by the stakeholders. Confidentiality, integrity, and availability acronym as CIA, are the characteristics that need to be maintained to provide security. However, availability added the most value in operational technologies, followed by integrity, with confidentiality being usually the last concern, eventually leads to the acronym AIC (also recognized as the security triad).
- Reliability: reliability is the capacity of a device or system to execute its necessary tasks for a given period of time, under certain requirements. The reliability of the system can be enhanced by analyzing which reliability facets an attacker might influence and implement the system and its security to overcome those attacks.
- Privacy: privacy is an individual or group's right to control what information relating to it may be retrieved, analyzed, and processed and by whom and to whom it may be revealed. Privacy relies on whether the stakeholders intend the information to be secure or restricted from other purposes, which are legally necessary. It is important to remain up to date with laws and guidelines, such as the EU General Data Protection Regulation (GDPR).
- Resilience: resilience is the evolving ability of a system that acts in a way that prevents, absorbs, and handles dynamic adversarial situations while executing the allocated tasks, and reconstitutes the operational functionalities after causality. When a single function fails, it does not force other functions to fail, so alternative forms to execute the failed task in the system should be available which can be executed instantly, quickly, and consistently.

## 8. Reflections on the Synthesized High Level Architecture

The previous section describes a synthesized high-level IIoT architecture to address the current challenges in the mining industry. A few research efforts tried to solve specific problems by IoT in the mining industry such as ventilation monitoring, accident analysis, fleet and personnel management, tailing dam monitoring, and pre-alarm systems. However, these individual IoT solutions created a challenge of interoperability between various solutions. There is a need for an IIoT architecture for the mining industry that follows the industrial guidelines defined by standard bodies and adopted by a large number of vendors.

Despite the fact that there exist several IIoT standards and initiatives which give guidelines to improve productivity in various industries. Unfortunately, the adoption of these standards practices in the mining industry is not straightforward because of the complex nature of the mining operations. The mining industry has unique challenges than other industries due to the infrastructural limitations at the mine sites. Mine sites can be of various types such as underground or surface. Those different mine sites deploy a different set of commercial devices and systems according to operational needs

which cause vertical fragmentation, hence the data transfer and communication between various systems and devices within a mine or among several mines become challenging.

This work studies the mining industry and related IIoT standards to synthesized a high-level IIoT architecture shown in Figure 4 which can benefit the mining industry. The layered architecture addresses the current challenges in the mining industry by offering a remotely controlled, automated, and interoperable environment to improve communication, data access, and data management. The architecture considers each mine site as an IIoT edge, which addresses all serious and complex issues by local edge gateway, name a few as asset management, monitoring and diagnosis, provisioning and deployment, and interoperability between different systems and devices. The information domain facilitates to improve data access by different systems, departments, and stakeholders. On top of the information domain, analytics performs fast and efficient data analysis to help determine business decisions to improve productivity as well as operational decisions at the edge to be aware of various risks. The predictability of mining operations and potential risks in the mining industry can also be achieved by analytics. Both information and analytics domains can be on-site within the mine or off-site in the cloud accessible by all departments and stakeholders securely. This IIoT architecture can also make mining companies smarter and more productive while enhancing the logistics processes and better customer relations by business application.

Moreover, the crosscutting functions describe in the architecture address many other challenges in the mining industry such as security, data distribution among different departments and stakeholders, possession of data, safety, privacy, and resiliency which are missing in the literature and other related work. The adoption of such IIoT architecture in the mining industry offers safer mine site for workers, predictable mining operations, advanced automation and operations of machines, interoperable environment for both traditional and modern systems and devices, automation to reduce human intervention, improving efficiency, ensuring worker and equipment safety and visibility, decreasing operational costs, reducing energy expenditure, and enables underground surveillance by converging operational technology (OT) and information technology (IT).

The resulted synthesized high-level IIoT architecture addresses the identified challenges in the mining industry. When starting new mine site operations, this architecture should be possible to adopt. However, there are impediments for adoption in ongoing mining operations since they are continuous and pauses cannot be afforded. It is also difficult to seamlessly change the operational and information technologies that are currently in action. Furthermore, introducing IIoT in the mining industry brings many new challenges as described in the next Section 9.

## 9. Open Research Challenges and Directions

This section discusses the open challenges with the purpose to provide some research directions in the domain of Industrial IoT. The enabling technologies for IoT such as wireless sensor networks, cloud computing, big data analytics, communication protocols, and embedded systems have many challenges that are addressed continuously in academia and industry. In contrast with IoT, IIoT is the convergence of IT and OT in industries and introducing various new challenges at the edge level. The edge in the mining industry is span across many mine sites, mills, and stockpiles with the presence of various different devices, systems, and communication channels. So the main challenges are in the edge of mine sites where the convergence of IT and OT took place and some of these challenges are as follows:

### 9.1. Interoperability and Integration

In this paper, we have described the two different modes to achieve interoperability between systems. The common-meta model is difficult to achieve because of legacy systems based on a different technological stack. Moreover, the main downfall of the broker-based approach is that if the number of interacting parties increases the need for broker implementation also increases and the overall complexity for this approach is $O(N^2)$. Where "N" is the number of interacting parties and the brokers

needed for the connectivity of these "N" nodes is $N(N-1)/2$. Moreover, integration is also a key challenge in the mining industry where it is really hard to achieve horizontal integration between departments of the mining industry such as exploration, operations, and management departments. The end-to-end integration between various stakeholders is also required. The mines are complex in nature where it is really difficult to deploy brokers and establish strong communication channels. There is ongoing research to solve the interoperability challenges in other industries [60], which can also facilitate to address the same problem in the mining industry.

### 9.2. Scalability

IIoT is supposed to pose several problems due to the possible unbounded amount of communicating actors. The worst case is that millions of possible event failures come with a wide scale of devices. Industrial systems must equip themselves with scalable infrastructures that are ready for expansion. In the future, many devices will be installed which will be mobile and constraint, so the IIoT solution needs to be adaptive and scalable to accommodate many devices. The mine sites are really big in the area and have many IoT devices deployed for various purposes. These devices and applications are mainly connected and controlled by some edge nodes. The modern mines are equipped with many robotic systems, autonomous vehicles, and drones for various operations. The edge of the mines should be able to scale up and down the number of devices and applications according to the needs e.g., there is a lot of material to take out for some reason from the mine so the edge node can scale the number of autonomous vehicles and robots to speed this process. The convergence of blockchain and edge computing can provide ways to address scalability challenges in IIoT [61].

### 9.3. Flexibility

As there are several IIoT applications, service provisioning to the various IIoT applications has been quite difficult according to their demands. IIoT consumers typically require on-the-move applications that are continuously optimized, personalized, value-added, and autonomous. The best way to design an IIoT solution effectively that expands and helps the organization to integrate different applications is to utilize a flexible architecture that will evolve into the future. In the context of the mining industry, the mine sites are dynamic in nature with various risks. For example, there is an alert for some accidents in the mine so the IoT devices and systems should be flexible to adapt the changes according to the needs. The manufacturers or the designers of IoT devices and applications need to develop flexibility into their products as the firmware updates not only facilitate customization upon initial deployment at a mine site but also allow new functions to be installed or update remotely without bringing them into in lab. A container-based flexible solution for industrial operations can address this issue such as described in [62] for industrial control applications.

### 9.4. Security and Safety

The complexity of IIoT technologies and the variability of IoT networking infrastructures contribute to a number of security problems. In the mine sites the control systems, IoT devices and applications need to be secure so that no malicious code can be injected to perform some dangerous tasks such as controlling the blasting system, vehicles, drones, and ventilation systems in the mines. Through IIoT it is necessary to have security in the bottom-to-top way which means that the IIoT application must adopt a safe booting process, firewalling, access control rules, device authentication, and able to perform security patches and updates. Moreover, the security verification and validation gives trust in the efficacy of the security checks developed to fix outlined security holes. It is a continuing task even though operational modifications have not been made to the system, which helps to not only keep the system secure from new attacks but also can investigate devices' behavior if malfunctioned to perform safety measures for the workers in the mines. Blockchain can be seen as security [63] and safety [64] enabler in IIoT.

## 9.5. Mobility Management

In terms of IIoT network and protocol, performance device mobility will pose different challenges. Mobility management is a difficult task in the industrial environment. Detection of device movement is important to be conscious of the device movement, which helps to connect the device to the different areas of the network. Moreover, there is a need to know the current position of the device in the network, which requires the exchange of control messages or signals. For example, an autonomous vehicle in a mine site can go to another mine site or travel between mines and stockpiles and need to be connected to edge nodes that are distributed across different areas. The mobility has two scopes, the local mobility scope within the local network and the global mobility scope. There are ongoing research activities for mobility management protocols to address various open challenges such as fault tolerance, balancing schemes, triangle routing, multi-homing, buffering techniques, and inter-mobility and intra-mobility cooperation [65].

## 9.6. Centralized to Distributed Systems of Systems

The current nature of IIoT architectures and solutions in the mining industry are mostly centralized, which collects data from every asset and store in the centralized cloud which results in a single point of failure and composability issues. The future trend is to go towards the distributed network which indeed more complex but is a long-term approach [66–68]. With the introduction of the distributed network in the mine sites, assets will become more responsible, there will be no single point of failure and it will encourage the open market where composability will no longer be a global issue. The distributed architecture can be composed of several independent systems that are connected through orchestration and late binding to form a system of systems. Independence brings benefits to the development, operations, and management of the different subsystems. This relates to the above-mentioned scalability and flexibility. The partly decentralized model of a system of systems relies on connectivity and networking and introduces a concept of core gateways or brokers to improve efficient interoperability and security. All these directions can be targets in future research.

## 9.7. Virtualization at the Edge

In the IIoT environment, like cloud platforms, edge also needs virtualization technologies for resource provisioning, task scheduling, and computation sandboxing. There are various virtualization techniques such as virtual machines, Linux containers, service mobility, and VM-based SDN [69]. Given the benefits of implementing the virtualization platform, edge computing has many characteristics that stick out from cloud computing, which raises different obstacles for virtual machines or containers management such as sparse distribution, limited resources, mobility of edge nodes, poor reliability, and limited service range. Moreover, the above-mentioned challenges can also be solved by achieving virtualization at the edge. The mining industry has many sites with different edge nodes and edge devices and there is a need to manage these nodes and devices at the edge similar to the nodes and applications in the clusters of cloud platforms. There are various projects to address this challenge such as OpenStack [70], KubeEdge [71], Edge virtualization engine (EVE) [72], OpenEdge [73]. These are currently not mature enough and require more research efforts.

## 9.8. Digital Twin

Apart from the above mentioned open research challenges, Digital Twin is another research direction for the mining industry. There are various definitions of digital twins and [74] concluded the definition as "a Digital Twin is a comprehensive digital representation of an individual object. It includes the properties, conditions, and behaviors of the real-life object through models and data. A Digital Twin is a set of realistic models that can simulate an object's behavior in the deployed environment. The Digital Twin represents and reflects its physical twin and remains its virtual counterpart across the object's entire life-cycle".

According to the recent survey [75], the digital twin has 12 main properties in the context of IoT such as representativeness and contextualization, reflection, replication, entanglement, persistency, memorization, composability, accountability/manageability, augmentation, ownership, servitization, and predictability. Whereas, some typical scenarios of digital twin usage are design and consolidation of products, prediction and simulation of the behavior, servitization of a physical product, and its augmentation.

In mining, the concept of digital twin provides a replica of a physical mine in a virtual environment to simulate and execute plans in the virtual model and helps to make value-driven decisions. It can help to conduct "what if" analysis to test various process methods such as blasting, crushing, and conveying to better understand the outcomes. Moreover, the simulation gives better insights to make it possible to predict and prevent failures of the assets in the mine sites.

## 10. Conclusions

The IIoT is a source and enabler of industrial automation, and opens possibilities for important insights applicable in several business areas. Together with Boliden we explore the mining life-cycle and analyze the current IT practices in the mining industry to identify various challenges which can address by applying a suitable IIoT architecture in the mining industry. We use the guidelines given by global IIoT standards and related initiatives to synthesize an IIoT architecture for the mining industry to address the existing challenges. The synthesized IIoT architecture can apply to bridge the technical gaps of interoperability and exchange of data that exist in the mining environment. The implementation of such high-level architecture can be made possible by considering various technologies such as OT/IT applications, IoT devices, cloud computing, edge computing, middlewares, big data, and business applications.

The article also listed significant open research issues and future directions for researchers. The open research challenges clearly depict that edge and fog computing is the most challenging layer for the mining industry and the same is true for various industrial sectors. The above-mentioned challenges for the edge can be addressed with the advancements in the virtualization techniques for edge computing. The virtualization-based solution in the edge computing for deployment, orchestration, updates, and upgrades are highly needed. Not only the mining industry but almost all the industries are quite complex at the edge with the advancement of IoT based micro-services, and systems. There are a great number of solutions available for cloud computing orchestration and deployment of services, but a standard or concrete solution targeting the same problem for the edge/fog does not exist.

**Author Contributions:** Conceptualization, A.A.; methodology, A.A; writing—original draft preparation, A.A.; writing—review and editing, O.S. and U.B.; visualization, A.B.; supervision, O.S. and U.B. All authors have read and agreed to the published version of the manuscript.

**Funding:** This research work has been funded by the Arrowhead Tools research project with Grant Agreement No. 826452.

**Acknowledgments:** We would like to thank Markus Frank, Daniel Lövgren, and Oskar Nilsson at Boliden AB and Sohail Manzoor at LTU for their valuable input and support in understanding the mining process and requirements on mining IT infrastructure.

**Conflicts of Interest:** The authors declare no conflict of interest.

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
