# Peer review of "A Study on Industrial IoT for the Mining Industry: Synthesized Architecture and Open Research Directions"

_2624-831X, doi:10.3390/iot1020029_

Round 1
Reviewer 1 Report
The paper deals with A Study on Industrial IoT for Mining Industry:
Synthesized Architecture and Open Research Directions
The paper is clear and well-structured.Goals are clearly presented.
Methodology needs to be clearly defined and improved.
Analysis has the appropriate range. Sufficient number of sources are used.
Website footnotes are absolutely insufficient - form. - e.g. cite https://productive40.eu/ is not clear. Web resources are missing in References
The paper lacks a bigger comparison with the results of already published works.
The paper should clearly define the difference between applications in the mining industry and other industries.
It is necessary to better define the mining industry - the type of coal, oil, stone, gas, etc.
The Open Research Challenges & Directions section is very well done. The conclusions are clear.
Reviewer 2 Report
Thank you very much for giving me the opportunity to review this paper that explores Industrial IoT for the mining industry.
In my opinion, the study can contribute to the IoT development discourse because it explores a particular industry, which is not prominent in the literature.
However, the study does not have a scientific structure that makes the narrative flow difficult to follow. Particularly, in the introduction, the study does not clearly illustrate the aim of the study and the literature gap that research found. Then, researchers introduced Boliden AB, but it is not clear why. The introduction needs also more references, especially in the last paragraphs.
The methodology part is lacked in the paper, so it is not possible to know whether this is a literature review or a systematic literature review. FOr this reason, it is not valuable to use espression like " No paper is describing n overall IIoT architecture..."
How can you prove it whether you do not conduct a systematic lt?
Moreover, the review includes only 26 papers that are not sufficient for a review article.
Section 8, which is the discussion part, requires more references in order to debate the previous studies and offer avenues of research.
Reviewer 3 Report
- Please clarify the limitations or problems of the related works and explain how you are solving them as a result of this study.
- Please be more clear about the requirements of Smart Mining.
- Describe how the architecture presented in this paper verifies efficiency, etc. in terms of smart mining requirements.
Round 2
Reviewer 2 Report
The novel version of the study has a better scientific structure which includes the methodology part. Authors better illustrate the method and why they include a Swedish company in the study.
To improve the study, I recommended including the following reference in lines 89-90:
Margherita, E.G., Braccini, A.M.: Industry 4.0 technologies in flexible manufacturing for sustainable organizational value: reflections from a multiple case study of italian manufacturers. Inf. Syst. Front. (2020).
Kiel, D., Muller, J.M., Arnold, C., Voigt, K.: Sustainable Industrial Value Creation: Benefits and Challenges of Industry 4.0. Int. J. Innov. Manag. 21, 1740015 (2017).
Benefits of IioT are not only related to the architecture of the system but also in economic, environmental, the social value which the technology generated.
- Another point is to improve is the limitation of the study and method.
